# Autonomous Temporal Pseudo-Labeling for Fish Detection

**Ricardo J. M. Veiga** 1,2,*, **Iñigo E. Ochoa** 3,4, **Adela Belackova** 3, **Luís Bentes** 3, **João P. Silva** 1,2, **Jorge Semião** 1,5, and **João M. F. Rodrigues** 1,2

1 Institute of Engineering (ISE), University of Algarve, 8005-139 Faro, Portugal; jpparente@ualg.pt (J.P.S.); jsemiao@ualg.pt (J.S.); jrodrig@ualg.pt (J.M.F.R.)
2 LARSyS, Institute for Systems and Robotics (ISR-Lisbon), 1049-001 Lisbon, Portugal
3 Centre of Marine Sciences (CCMAR), University of Algarve, 8005-139 Faro, Portugal; inigo.expositoochoa@imbrsea.eu (I.E.O.); abelackova@ualg.pt (A.B.); lbentes@ualg.pt (L.B.)
4 Marine Biology Research Group, Ghent University, Krijgslaan 281/S8, B-9000 Ghent, Belgium
5 INESC-ID, Instituto Superior Técnico, Universidade de Lisboa, 1000-029 Lisboa, Portugal
* Correspondence: rjveiga@ualg.pt

**Abstract:** The first major step in training an object detection model to different classes from the available datasets is the gathering of meaningful and properly annotated data. This recurring task will determine the length of any project, and, more importantly, the quality of the resulting models. This obstacle is amplified when the data available for the new classes are scarce or incompatible, as in the case of fish detection in the open sea. This issue was tackled using a mixed and reversed approach: a network is initiated with a noisy dataset of the same species as our classes (fish), although in different scenarios and conditions (fish from Australian marine fauna), and we gathered the target footage (fish from Portuguese marine fauna; Atlantic Ocean) for the application without annotations. Using the temporal information of the detected objects and augmented techniques during later training, it was possible to generate highly accurate labels from our targeted footage. Furthermore, the data selection method retained the samples of each unique situation, filtering repetitive data, which would bias the training process. The obtained results validate the proposed method of automating the labeling processing, resorting directly to the final application as the source of training data. The presented method achieved a mean average precision of 93.11% on our own data, and 73.61% on unseen data, an increase of 24.65% and 25.53% over the baseline of the noisy dataset, respectively.

**Keywords:** environmental monitoring; marine fishes; object detection; fish detection; pseudo-labeling; underwater video; deep learning

## 1. Introduction

The oceans are a vast and complex system, continuously changing and adapting due to external influences, and yet it remains ruled by equilibrium. Monitoring these changes in the underwater realms differs deeply from the surface environments, where the open space allows monitoring from the orbit of our planet using satellites, or using, e.g., IoT solutions.

These inherent constraints, intertwined with the constant underwater environmental shifting, create challenging obstacles to the mission of monitoring our oceans in real time. Because of our neglect towards a balance between exploration, maintenance of the oceans, and care for the environment, some ocean properties are changing, including the rising sea levels [1]. Our drive for knowledge of marine realms is magnified by its necessity [2].

Project KTTSeaDrones focuses on this necessity for deeper knowledge. In a nutshell (see more details in Section 3), one of the goals is to analyze the local marine environment by developing an underwater monitoring station prototype, which shares two main functionalities: (a) monitoring the coastal marine fauna and (b) station scalability, i.e., the station, when deployed in the ocean, should adapt itself (with no human intervention) to different environmental areas, conditions, and fauna, in such a way to work as a multipurpose underwater station.

Usually, surveys of the marine fauna are performed either by scientists or by local fisheries, normally resorting to trawls. The latter, although manually performed, offers daily updates on the state of the local coast; however, it is a biased method that relies heavily on the location of the capture and does not account for "undesired" species. Scientists focus on a broader approach that, while still performed by invasive techniques, present a better understanding of the state of the local marine fauna, although the obtained analytics were shifted in time to the date of the capture.

With the evolution of technology, the initial method of survey, which requires scientists to dive and visually analyze the site, can be replaced by baited remote underwater video (BRUV). BRUV allows a more reliable analysis in exchange for hours of manual footage examination performed by marine biologists [3]. This survey analysis is a tedious and repetitive process that produces variable outcomes and human errors correlated to the level of knowledge, attention, and vigor of the human specialist interpreting the data. The next step in the evolution is the necessity of relieving the burden of manually reviewing footage and developing automated processes to monitor the local marine fauna.

The present state of the art in machine learning, including deep learning (DL) techniques, allows for pursuing challenges not yet completely addressed, such as the one presented, using computer vision (CV) techniques for, e.g., surveillance analysis, which has been revolutionized by numerous DL algorithms, in particular by the You Only Look Once (YOLO) architecture [4,5] for object detection and classification.

In the field of underwater monitoring, the application of DL pipelines conjoined with CV is no novelty, with multiple authors already proving the success of these techniques [6–9]. Nevertheless, combining DL and CV approaches to analyze data from a single location or area results in custom models that are over-achievers to their specific trained domain. Monitoring stations typically are static by their nature and coastal surveys are normally performed along a specific range of coasts; therefore, local object detection and classification models for either real-time monitoring feedback or for automating the analysis of the obtained footage from the BRUVs perform extremely well on their data. However, this narrow approach introduces limitations in terms of scalability, where the previously trained models can outperform depending on environmental factors or different marine fauna. Therefore, if the manual creation of a custom dataset was previously required, it can become a recurring task that is necessary for the deployment of new monitoring stations or the evaluation of footage from disparate areas.

This paper focuses on the challenge of automation of the data annotation process [10] required for the training pipeline of an object detection model intended for underwater fish monitoring. Similar to previous work on fish detection [11], the detection stage is separated from the classification phase, with the latter being out of the scope of the present paper. The proposed approach tackles the creation of a new custom object detection dataset in reverse, focusing on the static nature of the target location footage of underwater monitoring stations and BRUVs, allowing the dataset annotations retrieved by automatic pseudo-labeling to refit a previously trained model from a foreign coastal environment to different marine fauna. The OzFish dataset [12] is used for model initialization through the technique of transfer learning [13], and the object detector process is performed using the YOLO architecture [5,14].

The main contribution of this paper is twofold: (a) The presentation of a transversal method able to automatically generate localized underwater datasets from foreign locations, regardless of the deployment application and the baseline dataset quality, type of species, and environmental conditions. (b) The presentation of a full pipeline, which allows the duplication of the method with different objects and environments (not limited to underwater objects—fish).

The structure of this paper is as follows: The present section introduced the agenda and the contributions of the paper; related works and the current state of the art are presented in Section 2, followed by a brief introduction to the KTTSeaDrones project in Section 3, and an analysis of the target data for this project and the pertinent available data

in Section 4. Next, the proposed method is presented and detailed in Section 5, succeeded by the experimentation and achieved results, accompanied by the discussion in Section 6, and finally, the conclusions and prospects for future work in Section 7.

## 2. Related Work

The first records of the use of video underwater date back to the 1950s, with an increased use of remote solutions during the 2000s [6], removing or reducing the presence of humans underwater. Regardless of the visual limitations introduced by the nature of our oceans, the use of underwater footage for marine monitoring, either streaming or recording, remains a viable and necessary solution.

Recently, the conjunction of a technological evolution marked the beginning of a new age in artificial intelligence (AI) techniques: the maturation of image sensors offer higher resolutions, sharper images, and increased dynamic ranges, which are essential to low-light applications, such as in underwater environments; the exponential expansion of computational power available, in particular the parallelization ability of graphical processing units (GPU), and the development of advanced mathematical libraries that revolutionized modern science and research, simplifying complex tasks while offering the ability to solve advanced challenges. The synergy of these technological advances converges in the separation of traditional and modern CV [7,15], introducing an expeditious development in underwater monitoring applications.

Considering the diversified nature of marine environments, the application of traditional CV proved to be convoluted and impractical, requiring static and adequate image conditions to perform classical CV algorithms [16]. With the emergence of DL techniques and their inherent versatility and performance, multiple complex tasks were finally efficiently achieved, replacing the previous state of the art and swiftly approaching metrics similar to human performance—even surpassing it [17,18]. However, the CV object detection performance accomplishments, even while achieving exceptional generalization, endure different limitations and variability while performing underwater [19]. Nevertheless, while the creation of a universal marine classifier and object detector represents colossal challenges, narrow applications of DL methods to monitor local marine fauna proved to be successful [7].

A crucial requirement for the application of DL methods is data; specifically, properly annotated datasets [20]. Subsequently, the performance of any trained DL model is directly related to the chosen input data, either by its quality or quantity. Notwithstanding the progress achieved through preprocessing procedures, data augmentation, tuning hyperparameters, or fine-tuning convolutional neural networks, the base outcome remains directly correlated to the trained dataset [21]. Properly annotated data results from the combination of dedication, skill, and time. The initial approach to data gathering bifurcates into a question of generalization or applicability. Therefore, during the first stage of a custom object detection project, the abstractly chosen data, prior to the labeling phase, can already influence and bias the output model [21]. However, distinct applications may require generalization, while static underwater monitoring stations may benefit from fitted models.

Underwater environmental biodiversity raises a convoluted obstacle to the generation of a universal dataset to observe and analyze marine ecosystems. Therefore, available underwater visual datasets [2], either for classification or object detection, normally represent local marine visual conditions, restricting an attainable application to dissimilar underwater areas.

A landmark underwater dataset aimed at the study of marine ecosystems is the Fish4Knowledge dataset (F4K) [22], was obtained from the capture of live video in the open sea. The ImageCLEF initiative introduced the LifeCLEF14 and LifeCLEF15 datasets [23], increasing the number of annotated videos containing fish, while also resorting to the F4K dataset. Focusing on the task of fish recognition, the WildFish and WildFish++ datasets [24,25] made available hundreds of thousands of images containing thousands of fish species. The availability of the OzFish dataset [12] increased the published videos and

annotated bounding boxes to the order of thousands. Additional segmentation annotations were presented on the DeepFish dataset [26], which also introduced marine videos under different conditions. The SUIM dataset [27] focused on the semantic segmentation of underwater images, while the NorFisk dataset [28] focused on fish farms.

Current DL object detection methods are divided into two main approaches, the one-stage detectors [5,14] and the two-stages detectors [29,30]. The idea behind two-stage detectors is simple: the first stage extracts sparse region proposals through a region proposal network (RPN), which are later classified, and the corresponding bounding boxes are found through regression tasks in the second stage. The most relevant two-stage detectors are the R-CNN [31], SPPNet [32], Fast R-CNN [33], Faster R-CNN [29], and Feature Pyramid Networks (FPN) [30]. One-stage detectors remove the region proposal step by proposing predicted bounding boxes directly from the input images, therefore increasing computational efficiency. Prominent one-stage detectors are the SSD [34], RetinaNet [35], and the YOLO family [4,5,14,36].

In summary, two-stage detectors are more accurate in exchange for computational performance, while the one-stage are more computationally efficient in lieu of accuracy. Therefore, the choice between both approaches is defined by the desired applicability for the resulting models, either for real-time applications, such as underwater monitoring stations, or for more precise demanding deployments, such as the medical imaging field.

YOLOv4 [5] solidified the importance of data augmentation and post-processing techniques, while also promoting the evaluation and integration of techniques introduced at first with two-stage detectors, such as the FPNs [30]. Notwithstanding the current evolution and performance of modern object detection algorithms, the diverse ever-changing nature of underwater environments introduces new challenges, either in the input stage, requiring image enhancement techniques due to inherent light propagation limitations, or at the neck, due to multiscale-targeted objects, which may camouflage with the environment, or directly at the chosen dataset, which may prove unreliable for different locations or conditions.

Multiple methods have been investigated to detect, classify, or perform both when monitoring fish. Regarding the dataset used, there are two distinguished approaches: the integration and evaluation of previously published datasets, and the generation of a custom dataset for evaluation of the published methods. Cutter et al. [37] published the Labeled Fishes in the Wild dataset and applied several Haar cascade classifiers [38], obtaining precise detection rates between 66% and 81%. Choi [39] used a custom dataset for comparison between traditional detection methods through the Histogram of Oriented Gradients (HOG) [40] feature descriptor plus support vector machine (SVM)- [41] and DL-based detection methods, resorting to the GoogLeNet [42] network architecture, reporting F-measure results always below 49% for the HOG+SVM approach versus a minimum of 55% for the GoogLeNet DL approach. Choi [39] fused traditional background subtraction methods for fish detection while fine-tuning a pre-trained GoogLeNet [42] network for fish identification on the LifeCLEF15 dataset [23], obtaining counting scores above 85% and precision scores above 71%. Li et al. [43] and Ash et al. [44] applied Fast R-CNN and Faster R-CNN on the same dataset, achieving a mean average precision (mAP) of 81.4% and 82.7%, respectively.

Qin et al. [45] proposed the extraction of the foreground, resorting to sparse and low-rank matrix decomposition, later applying a linear SVM for classification, achieving an accuracy of 98.64% on the F4K dataset classification task. Mandal et al. [46] combined a Faster R-CNN with three classification networks, namely two modified AlexNet [47] networks and a VGG-16 [48] network, obtaining an average precision of 82.4% on the classification of their custom dataset. Xu and Matzner [49] employed the YOLO architecture on a challenging underwater custom dataset, obtaining a direct mean average precision of 53.92%, proving the challenging variability of the object detection task underwater.

Salman et al. [50] evaluated the published findings with the LifeCLEF15 dataset, and additionally with the F4K dataset [22], exploring the temporal information through

Gaussian mixture models (GMM) and optical flow, and a CNN to train an R-CNN network, achieving an F-score, i.e., the weighted harmonic mean between precision and recall, of 87.44% and 80.02%, accordingly. Jalal et al. [51] also evaluated the published findings with the LifeCLEF15 dataset and additionally a custom dataset; similarly, the temporal information used the previously proposed methods, although a parallel YOLO network was applied, combining the proposed location and classification at a later stage, increasing the F-score on the LifeCLEF15 to 95.47%, and achieving a classification accuracy of 91.64%, while obtaining 91.2% detection accuracy on the custom dataset, and 79.8% on classification.

Pedersen et al. [52] published the Brackish dataset, comparing the performance of the YOLOv2 and YOLOv3 through the Intersection over Union (IoU) metric, achieving with the latter a $mAP_{IoU=0.50}$ of 83.72%, and a more challenging $mAP_{IoU=0.50:0.05:0.95}$ of 39.83%. Zhang et al. [19] also evaluated the previous dataset with the YOLOv4 [5] network, the tiny YOLOv4 [53] architecture, and their proposed YOLOv4 modified network with MobileNetv2 [54] as the backbone, achieving a mAP of 93.56%, 80.16%, and 92.65%, accordingly, while reducing the necessary model parameters.

Labao and Naval Jr [55] suggest an ensemble of two-stage object detection networks connected by long short-term memory networks through cascade structures applied to a custom challenging dataset containing variable quantities of small-scale objects, obtaining an average of 60% on both precision and recall metrics. Ditria et al. [18] proved the reliability of DL approaches for underwater monitoring, comparing the results obtained from the implementation of a Mask R-CNN [56] network on a custom dataset against an analysis from marine experts and citizen scientists, recording an increase in the abundance detection of 7.1% and 13.4% for single images, and 1.5% and 7.8% for videos, respectively. Knausgård et al. [11] used the F4K dataset to train a pre-trained classifier and further train with a limited custom dataset. The detection and classification tasks were divided, with the YOLOv3 [36] architecture responsible for the object detection stage and a Squeeze-and-Excitation [57] architecture employed for classification. A mAP of 86.96% was obtained for the object detection task, while the classification achieved an accuracy of 99.27% on the F4K dataset and 83.68% on the custom dataset.

Stavelin et al. [9] also implemented the YOLOv3 on a limited annotated dataset, retaining the detections above 25% confidence score and manually correcting the predictions for further network retraining, achieving a mAP of 88.09%. Ditria et al. [58] created combinations of five diverse custom datasets obtained from reef and seagrass footage from different locations, and trained an implementation of Mask R-CNN to demonstrate the improved generalization across different habitats.

As it can be seen, previous works have approached the integration of DL-based object recognition techniques for autonomous underwater monitoring by either focusing on improving state-of-the-art results across public datasets, implementing novelty methods to underwater conditions and comparing performance evolutions, or building localized custom datasets. More importantly, the majority of recently presented works have increasingly required the creation of new and locally fitted datasets, normally distributed for only one location per dataset. None of the authors presented a solution that allows an autonomous adaptation to different underwater locations by automatically generating human-free distilled pseudo-labels, promoting a faster and easier expansion in underwater monitoring, independent of the geographic location.

## 3. Project KTTSeaDrones

The multiple effects of climate change across the previous decades and the rise of the ocean's temperature introduced detrimental consequences to coastal marine life, such as the arrival of invasive species to the shores [59]. It has become imperative to access real-time data, providing our oceanographers and marine biologists with the necessary information for the formulation of real-time solutions. The KTTSeaDrones project aims, among other goals, at installing a state-of-the-art underwater monitoring station for autonomous reports of the local fauna, in terms of the quantity of local species and the presence of invasive

species at multiple locations. The station should work under variable environmental conditions and must be equipped with edge processing (local) and offer the possibility of cloud or hybrid processing. The station has a system (camera) for fish detection and recognition, and the quantification of species, and should generalize across the regional underwater fauna and environmental conditions, becoming also the baseline model for regional baited remote underwater videos and future monitoring stations.

The generic block diagram for the KTTSeaDrones underwater monitoring station prototype is presented in Figure 1. The diagram shows on the left the Land Station (LS), composed of power and data units that are connected to the Underwater Monitoring System (UMS), on the right, by submarine optic fiber and power cables. This system (UMS) can work in a stand-alone mode or in a hybrid mode, communicating with LS at specific hours of the day, or in a fully connected mode, transmitting in real time all the information that is acquired by the sensors to the LS. The underwater station uses several sensors to acquire information, such as sonar, acoustic, environmental, and camera, etc. Additionally, included are the edge processing units, where all the relevant processing is performed. All the data is stored in the data module (and transferred, when and if requested, to the land station). More details about the complete project and the different modules can be consulted on the project website at https://kttseadrones.wixsite.com/kttseadrones (accessed on 1 May 2022).

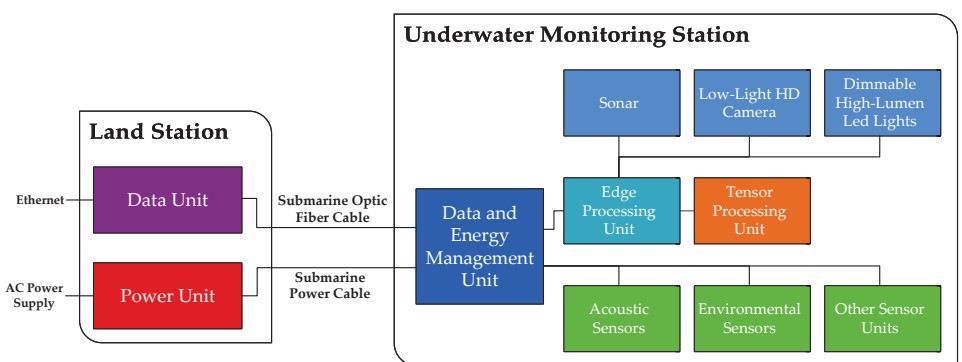

**Figure 1.** Diagram presenting the overall structure of the underwater monitoring station prototype, with the underwater monitoring station connected to the land station by submarine optic fiber and power cables.

## 4. Data

As already mentioned, datasets are a crucial part of the modern DL object detection method, where the quality of the annotations, scenes, and object variability, as well as the appropriate and balanced quantity of samples, are pivotal for the training of unbiased and functional models. This direct correlation can be observed between the evolution of two landmark large object detection datasets, the PASCAL Visual Object Classes Challenge [60], and the MS-COCO dataset [61], with the latter increasing the number of available images, annotations, classes, and, more importantly, the presence and density of small objects, approaching the object distributions closer to the real world [17]. Therefore, the data collected and annotated should reflect similar guidelines, either in location precision or object density, to maintain a uniform level of labeling quality.

The demand for creating custom datasets emerges from the absence of the targeted classes on available datasets, the unreliability of noisy datasets, and performance improvement limitations caused by constrained localization or for specific controlled applications. However, the acquisition and proper labeling of data for the generation of a reliable dataset is a tedious and time-consuming process, which is especially dependent on the annotator's skills, attention, and endurance. Nevertheless, several approaches were proposed to expedite the annotation process, focusing primarily on expanding limited annotated data to automatically annotate unlabeled data, i.e., pseudo-labeling.

Regarding object detection, three mechanisms are usually chosen: a semi-supervised learning approach, exploring the relation between labeled and unlabeled data; an active learning approach, which automatically selects the next most relevant unlabeled sample, optimizing the annotation process; and the automatic generation of a dataset based on a pre-trained model, which relies profoundly on the ability to properly generalize unseen data. Nonetheless, these methods are primarily suitable for the classification task, while still empirical for the object detection task.

In order to test the model presented in this paper, underwater footage from the Algarve seabed in Portugal was gracefully provided by the Algarve Centre of Marine Sciences (CCMAR). This data was obtained using BRUV techniques, consisting of 21 uncut and unlabeled videos from different locations, as can be observed in Figure 2, containing diverse habitats and environmental conditions, which is either a generalization challenge or a baseline solution [58]. The gathered recordings have an average duration of 33 min each, summing to a total of 695 min, while still containing initial parts onboard the research vessels, and random endings, which are either underwater, transitioning, or back onboard. A collection of 383 unseen images from different areas were annotated by a marine biologist, providing a ground-truth test evaluation.

Figure 2 shows in the top row samples (frames) of CCMAR unlabeled footage in different environments, in an *underwater stage*. This targets the sector of the video where the camera is stable on the seabed. In the middle row, it is the same as previously, but with the ground-truth annotations performed by a marine biologist. In the bottom row, *negative sample* frames are obtained from the video that do not occur underwater, i.e., the *overwater stage*, where the camera is out of the ocean, for the two leftmost images, and the *transition stage* for the two rightmost images. The transition state focuses on the phase where the camera is being dropped or pulled onboard. This process of overwater and transition footage is completely automated, but it is out of the focus of the present paper. A following paper will detail all the information on the CCMAR dataset, which will be publicly available, including the ground-truth annotated by a marine biologist, the classification of frames that are overwater, in transition, and underwater, as well as the resulting labeling provided by the present method.

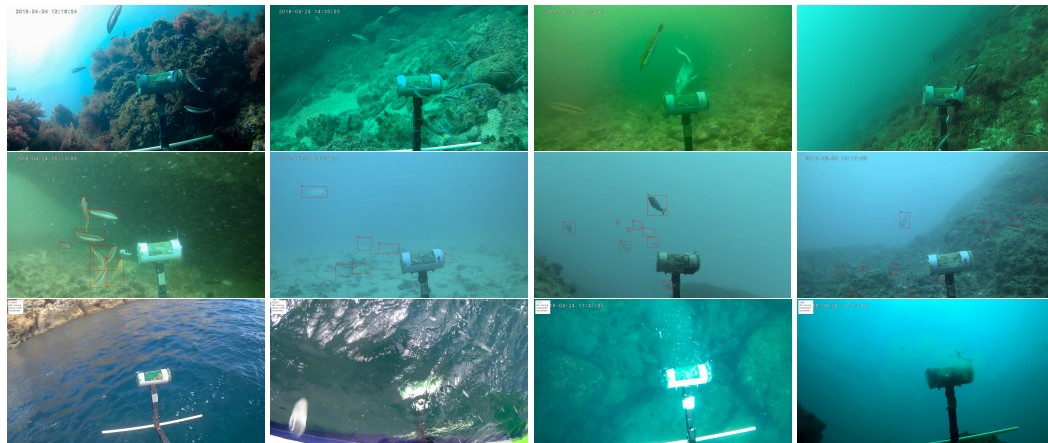

**Figure 2.** Top row: sample of CCMAR unlabeled footage. Middle row: ground-truth annotations of CCMAR videos. Bottom row: negative sample images obtained from the video classification step.

Finally, it is important to stress that for the initial transfer learning process applied in the method (see Section 5), the OzFish dataset [12] was used, which is a BRUV public dataset of the Australian marine fauna. Although this data reflects antipode species to our targeted application, which would result in an unreliable object detection model, the use of a contrasting and noisy dataset is a fundamental part of our presented method to demonstrate the knowledge transferability and applicability in the generation of an automatically local labeled dataset.

The object detection fraction of OzFish was evaluated initially through an exploratory data analysis of the remaining 1753 frames; some images with annotations are presented on the top row in Figure 3. One important remark about this dataset is the outsourcing nature of the available annotations, which introduces noise to the dataset either by wrong or missing annotations or bounding box labels, e.g., the existence of 935 annotations with an area smaller than 1% of the mean bounding box area, including an annotation with an area of 9 pixels. Some examples can be observed in the bottom row of Figure 3. Albeit the existence of inconsistencies in the available data for the object detection task, as mentioned, such as wrong annotations, multiple bounding boxes for the same object, and miniature boxes with no recognizable target inside, we decided against cleaning and preparing the data for a traditional object detection model training pipeline, focusing on automating the learning stage to filter and minimize the effects of noisy data on the training process instead of filtering the data a priori. Furthermore, employing the method on noisy data without the curating step significantly reduces the overall time spent on an object detection project. Nevertheless, although part of the data is unreliable, the remaining data still contain relevant information and should not be discarded.

Figure 3 shows in the top row samples of OzFish dataset; in the bottom row, examples of erroneous annotation; from left to right, overlapped annotations over the same target, and a negligible small bounding box, reef, feeder structure, starfish, and floating rope, all annotated as fish. In the next section, the developed method is going to be presented.

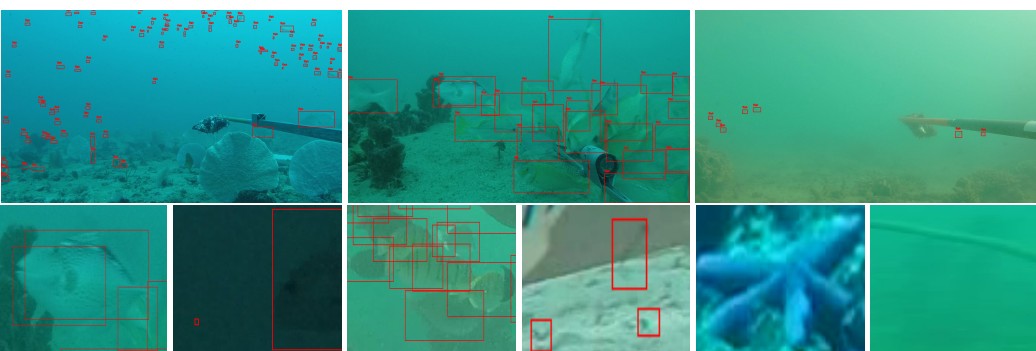

**Figure 3.** Top row: sample of OzFish dataset object detection images. Bottom row: examples of wrong annotations. From left to right: overlapped annotations over the same target, negligible small bounding box, reef, feeder structure, starfish, and floating rope, all annotated as fish.

## 5. Method

Beforefurther explanation of the method, it is important to stress that it is only applied in frames from the *underwater stage* (see Section 4). The method follows the overall pipeline presented in Figure 4, and it is divided into five main modules. It begins with the (i) *Data Preparation* pipeline ((a) to (f)), which consists the data generated for the initial transfer learning process. This data comes from two sources: CCMAR *negative samples* and the OzFish dataset (see Section 4). Then, applied is the (ii) *Transfer Learning* pipeline (g), which consists preparing the baseline model training. This starts a training loop cycle, which is the (iii) *Self-Supervised Learning* pipeline ((h) to (k)), where data mining is applied to the CCMAR data, collecting multiple annotations, which are then used for pseudo-labeling, distilling, and preparing the new images to continue training the previous models, returning then to the video data mining stage, and only exiting to the (iv) *Prepare Data for Final Model* pipeline ((l) to (m)), which generates a final dataset after stalling the training loop. Lastly, the pseudo-labeled dataset is generated. The dataset is then used for the (v) *Final Model* pipeline (n), which is finally adapted to local fauna (in this case Portugal—the Algarve coast) and consists in training a new and localized (final) model. The entire pipeline is presented in detail below.

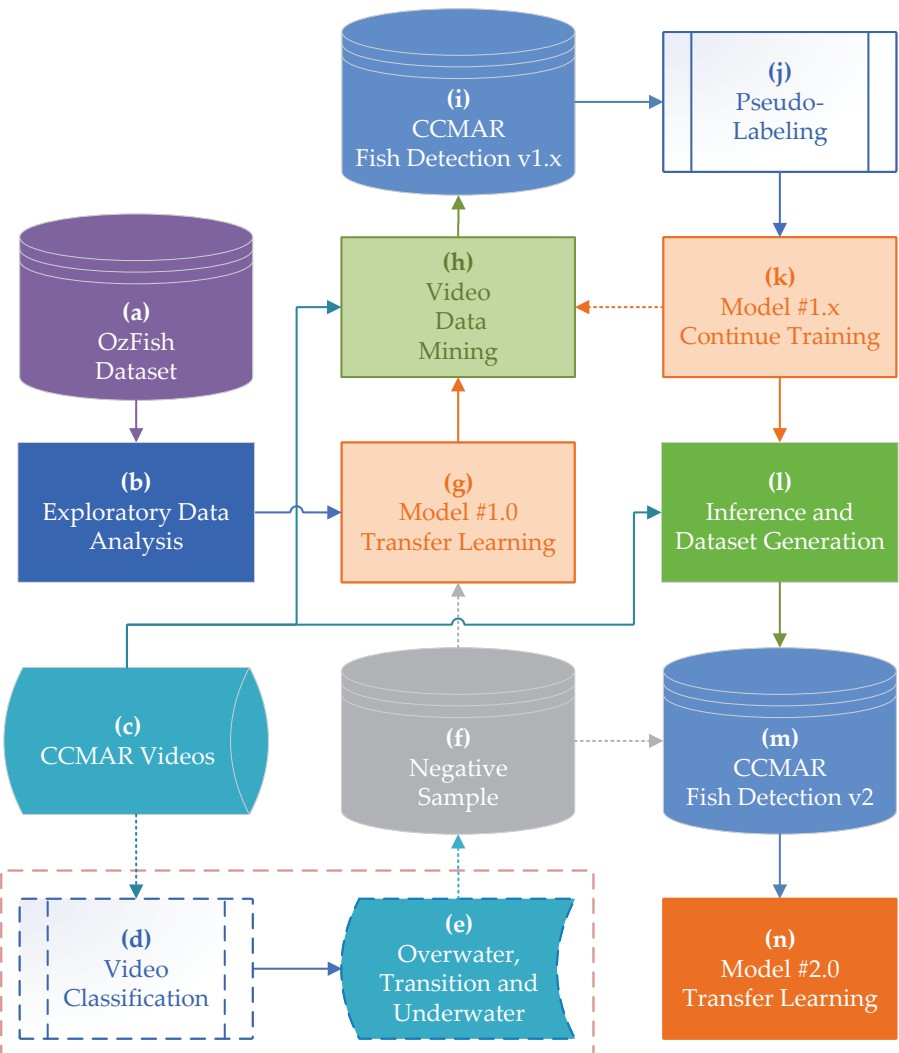

**Figure 4.** Overall pipeline of the proposed method. Each stage is identified by (∗).

### 5.1. Data Preparation

Pipelines (a) to (f), according to Figure 4, are comprised of data preparation to start the method. Pipelines (a) and (b) consist in obtaining OzFish dataset [12] and performing exploratory data analysis (EDA). Regardless of using all the available annotations, the (EDA) remains an important step in abstractly evaluating and understanding the data present in any dataset, particularly in the visual assessment of the data viability for the intended detection and recognition targets. In parallel, in (c–f) the CCMAR videos were accessed and segmented into *overwater*, *transition*, and *underwater* (see also Section 4). After this classification, the *negative samples* (f) are used in conjunction with OzFish data for step (g).

### 5.2. Transfer Learning

The YOLOv4 architecture [5] was adopted for the implementation of pipeline (g). The main advantage of using the YOLO architecture is the balance between its fast detection rate, by design, and the high accuracy maintained, even across complex scenes, while requiring moderate computational power. Furthermore, since the whole image is considered either during the training or inference phases, the additional contextual information contributes to a lower false-positive rate, although, due to the removal of the region proposal step, some objects, specifically smaller objects, can fail to be localized [62]. It is important to stress at this moment that the YOLOv4 is used across the different steps of the method, enabling

the possibility of the detection of multiple planes of distance, extensive environmental and target generalization, various input image resolutions, and a further reduction of type I and II errors.

The baseline-trained model follows the traditional transfer learning object detection training algorithm, where the weights are already initialized from previous training, resorting to higher computational power requirements on a major carefully curated dataset. The transfer learning technique [13] grants the advantage of a faster model training, requiring fewer resources while achieving similar accuracy and performance results to the original training models. In the present case, the initial YOLOv4 network was previously trained on the Common Objects in Context (MS-COCO) dataset [61], which consists of 80 classes distributed through more than 118 k images for training, and 5 k images for validation. Arguably, the use of a multi-class network for the object detection task of detecting one class seems unreasonable, although the targeted class for the purpose—fish—contains multiple species, with distinct peculiarities across them. Furthermore, isolating the detection and classification stages aims at providing, at a later stage, a crucial element for the recognition of invasive fishing species.

The default training configuration of the baseline model using the YOLOv4 network architecture was altered to detect only a single class named "fish", and the input three color channel images remained at $608 \times 608$ pixels. The OzFish dataset images were divided into training, validation, and test subsets, following a ratio of distribution of 80%, 15%, and 5%, respectively, for a total of 1753 images. The CCMAR *negative sample* was added, proportional to the number of original images from the OzFish dataset, i.e., 1753 images were retrieved from the overwater and transition states of the original CCMAR videos.

Additional default hyperparameters were altered as follows: the number of anchor boxes remained the same, although their values were recalculated to improve small object detection, either on a further plane or on smaller species, replacing the initial MS-COCO dataset values by pairs of width and height as: $16 \times 25$, $37 \times 50$, $73 \times 72$, $61 \times 137$, $125 \times 119$, $123 \times 265$, $209 \times 184$, $276 \times 331$, and $483 \times 518$. The batch size was set as 128, with the subdivisions set at 16. Even though using a large batch size affects negatively the model generalization [63], it became necessary, as this noisy dataset, after multiple initial training, was unable to converge to the global minima. The maximum number of iterations was set to 10k, with steps of 8k and 9k; the learning rate was adjusted to 0.005, while the decay remained at 0.0005 and the momentum at 0.949. Furthermore, also applied, from the bag of freebies introduced with YOLOv4, were data augmentations techniques as follows: the mosaic augmentation, which generates four-image mosaics during training instead of using only single images, further improving the accuracy, and the blurring, which, although having not improved the accuracy at the experiments presented with YOLOv4 [5], helped reduce the rate of type-I and II errors resulting from wrong or missing annotations present on the OzFish dataset.

### 5.3. Self-Supervised Learning

Pipeline (h) to (k), starting with (g) as the baseline model (trained on the OzFish dataset and CCMAR *negative samples*), initiates a loop of self-supervised learning, consisting of four distinct stages in the pipeline (Figure 4): (h) CCMAR video data mining; (i) the generation and aggregation of a temporary dataset; (j) a pseudo-labeling process over the filtered frames, and (k) continual object detection model training.

The data mining stage (h) essentially performs inference using the current model version (g) over the *underwater* frames of the CCMAR videos, while resorting to a simpler tracking method using the Euclidean distance, which associates nearby bounding boxes coordinates across multiple frames to maximize and fill parts of missing annotations across those multiple frames. This tracking process is performed forwards and backwards, allowing for a more complete analysis of the available footage, i.e., tracking the original path of a detected target. For example, a target that originated from a background area and was briefly detected with low confidence, thus discarded as a false positive, may have

a possible path retrieved for it when the target is continuously detected on a foreground plane. Associating the additional information from the tracking method, even with the limited propagation across multiple frames, with the bounding boxes detected running inference through the current model, enhances the confidence of true-positive detections while assisting in the process of discarding false-positive detections.

The result of the previous step is a collection of annotations paired with their confidence levels for each frame (i). Considering this, all the current information for training a new model or continuing the training of the previous model would exacerbate an already noisy model. Therefore, it was decided in this stage which frames and annotations would be considered for the next step. This process is achieved by performing autonomous EDA on the obtained results, isolating the clustered detections, prioritizing the detections by their hybrid confidence level and their inference plus tracking information, and defining a temporal threshold, $t_t$, for the chosen frame, in order to prevent over-similar images from the same sequences, and thereby improving the dataset generalization. A selection is filtered from this analysis considering all the scattered detection locations based on: (i) the number of detections by image $n_{di}$, and the confidence level of the detected bounding boxes, $c_l$ (with, in the present case, $t_t = 20s$, $n_{di_{present}} >= n_{di_{past}} + 1$, and $c_l >= 70 \frown 90\%$). An example of the evolution of the model predictions during this loop stage can be observed at Figure 5.

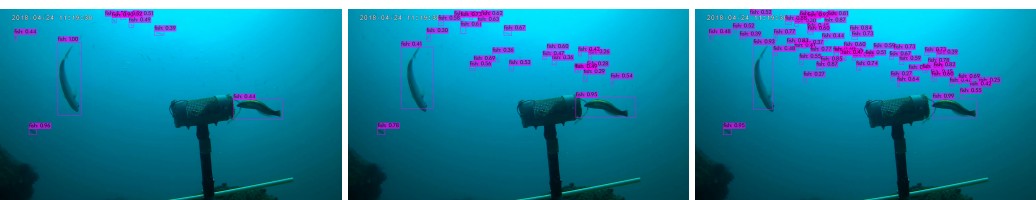

**Figure 5.** Example of model progression during the self-supervised pseudo-labeling loop cycle (see text).

The following step could define the entire loop. The (j) pseudo-labeling method is an unsupervised learning technique, where models train on a small labeled subset of the dataset and expand the labels across the unlabeled data by initially calculating the loss on the labeled data, followed by the prediction of pseudo-labels over the unlabeled data, and finally, the calculation of the loss of the unlabeled data. In comparison, the initial model training was performed on foreign species data, related only by association of belonging to the same animal class. Therefore, the automated labeling method presented initiates using unlabeled sequences of frames from the underwater portions of the CCMAR footage. This stage is more prominent after additional loop cycles, when the previous pseudo-generated labels are used to expand the bounding boxes pool or to increase confidence.

The obtained pseudo-data is then filtered by high-accuracy labels: above 90% accuracy and decreasing on each loop until 70%, with multiple detections across frames, i.e., tracked fish with more than 20 highly accurate detections; a selection of frames containing the higher amount of detections in comparison to other frames within the same loop during small sequence windows of 20 seconds; and prioritizing environmental diversity and conditions, i.e., footage from different locations. The use of densely filtered annotations of the CCMAR footage to (k) continue the training of the model emphasizes the importance of understanding that, in an object detection architecture, in this case YOLOv4, while conducting the training stage on a one-class model, the training is actually being performed on two classes: the target and the background, or, technically, the training is always performed on the number of classes plus one.

Therefore, the background information is extremely valuable, especially when containing faulty annotations, either missing false negatives, or erroneous false positives. Considering that in the first-loop cycles only highly confident annotations are going to be considered for continual training, this leaves the remaining low-confident positive detections scattered across the images. To prevent the effects of this noisy information during the

training stage, a progressive averaging blur was applied over the undetected areas, which consists in increasing the box filter kernel, from $3 \times 3$ until $21 \times 21$, with the distance from the detected areas.

The training configuration during the self-supervised learning process differs from the previous baseline configuration, with the input images resolution increased to $960 \times 960$, in order to improve the detection of smaller objects, with the anchors being recalculated with each cycle to the current sub-sample used for continual training. The maximum number of iterations increased with the range of the new high-confidence images available and the batch size remained at 64, while the subdivisions were altered to 32. More importantly, the learning rate is altered between 0.005 and 0.001 through a direct correlation to the predicted accuracy rates.

This self-supervised learning loop continues (h–k), always using the latest continual-trained model to process and reanalyze the CCMAR underwater footage, until reaching a stalling threshold, where lowering the high-confidence pseudo-labeled detections from 90% until 70% fails to introduce significant new annotations when compared to the aggregations from the previous cycles, i.e., the amount of new annotations is lower than 10% of the total amount of accumulated annotations. Another essential highlight is the importance of attaching an object tracker to the presented method in order to improve the detected object confidence. This enhancement allows for the propagation of high-confidence detections of foreground objects across different environmental conditions, or while navigating across multiple depths of field. Due to the simplicity of the object tracker applied, this certainty required two conditions: separate pathways from different targets, especially if the cross is slow or stalled, and a minimum of a weighted average of confidence detections superior to the required threshold, with the minimum accuracy starting at 90% and decreasing until 70% with each loop, i.e., when a target is detected and tracked, allowing for a continual association of different precision metrics and prioritizing higher accuracy.

### 5.4. Data Preparation for Final Model

Pipelines (l) to (m), after reaching the stalling threshold, advance into the (l) generation of an (m) updated dataset, matching the number of extracted labeled images from the negative sample bundle, i.e., images not containing any objects similar to the intended targets. Albeit being an analogous process in comparison with the first stages of the self-learning loop, the compounded information obtained from the multiple loop cycles provides an additional filtering of images containing false positives, while also decreasing the occurrence of false negatives. During these steps it is possible to determine the amount, label locations, and type of annotated images required for further training, allowing an interactive interchange between the aimed applications for the future trained model: edge devices, mobile devices, or cloud processing, where the nature of the required images for training may shift according to the environmental conditions and targets.

### 5.5. Final Model

At the last stage (n), the previous model is finally used to continue training, or a new model is trained, by either randomized weights or using a transfer learning approach. The configuration used for the training of new models during this stage varies depending on the intended application and device. In the present paper, maintaining the continuity, the transfer learning approach was used again to achieve the final model.

It is important to stress once again that, despite using YOLOv4 as the final architecture, the automatically generated final dataset (Section 5.4) can be used to train simpler and more localized object detection networks, which can immensely reduce the necessary computational power on an edge device, while releasing resources that can further improve the detection rate and accuracy. The next section will present the tests performed with the methods and the results achieved.

## 6. Experimental Results and Discussion

The presented results were obtained using a single NVIDIA RTX 3090, with 24 GB of VRAM used for all the training of the presented method, with a main focus on the ability to reproduce similar results without the necessity of resorting to higher computational power. The presented method was also adapted and reproduced on an NVIDIA RTX 2070 Super with only 8 GB of VRAM, achieving similar results, thus verifying the versatility of this method for further adaptation and applicability.

For the baseline training stage of the presented method, we evaluated the OzFish dataset [12], bare and accompanied by the *negative sample*, i.e., images not containing any targets, therefore, without annotations. The results can be observed in Table 1, where the $mAP_{50}$, i.e., the mean Average Precision (mAP) at Intersection over Union (IoU) of 50%, increased by 4.47% from 68.46%, and 9.99% from 25.94%, for $mAP_{75}$, with the addition of a *negative sample*. The average IoU also increased by 7.68% from 59.28%, and interestingly, the recall, which is the metric responsible for measuring the number of targets identified in an image, only increased by 1% from 65%, while the precision, which evaluates if the detection is a true positive, increased by 7% from 79%. Therefore, the addition of negative samples, even from unrelated images, and without representing the dataset bare background, considerably improves the model accuracy, especially at a higher IoU, although it did not significantly improve the recall, i.e., the metric used for evaluating the number of positive detections for available objects in the image.

**Table 1.** Model training parameters and results using the YOLOv4 methods.

| Method | Dataset | Size | #Images [1] | #Annotations [1] | $mAP_{50}$ | $mAP_{75}$ | $\overline{IoU}$ [2] | Precision [2] | Recall [2] |
|---|---|---|---|---|---|---|---|---|---|
| YOLOv4 | OzFish | 608 | 1753 | 43,572 | 68.46% | 25.94% | 59.28% | 79% | 65% |
| YOLOv4 | OzFish+NS [3] | 608 | 3506 | 43,572 | 72.93% | 35.93% | 66.96% | 86% | 66% |
| YOLOv4 | CCMAR | 608 | 4100 | 13,318 | 88.33% | 65.78% | 70.36% | 85% | 84% |
| YOLOv4 | CCMAR | 608 | 12,500 | 67,750 | 92.99% | 74.49% | 73.44% | 87% | 86% |
| YOLOv4 | CCMAR | 416 | 35,510 | 113,886 | 91.90% | 68.96% | 73.31% | 88% | 83% |
| YOLOv4 | CCMAR | 512 | 35,510 | 113,886 | 92.74% | 72.67% | 74.28% | 88% | 84% |
| YOLOv4 | CCMAR | 608 | 35,510 | 113,886 | 93.11% | 74.70% | 75.10% | 89% | 84% |
| YOLOv4-tiny | CCMAR | 416 | 35,510 | 113,886 | 85.00% | 45.33% | 68.74% | 87% | 71% |
| YOLOv4-tiny-3l | CCMAR | 608 | 35,510 | 113,886 | 89.66% | 61.02% | 73.69% | 90% | 73% |

[1] # means number of. [2] For IoU threshold at 50%. Confidence threshold at 50%. [3] *Negative sample* images.

Measuring the model evolution during the self-supervised pseudo-labeling loop is a convoluted challenge, considering that the main objective for the proposed method is the autonomous generation of an object detection model based on unlabeled footage, from and for the local fish population from unlabeled footage: resorting to carefully annotated images for algorithmic guidance during the self-learning cycle would corrupt the purpose. Therefore, the results of the self-learning loop can only be genuinely calculated after the end of the loop cycle. However, it is possible to observe the evolution of the autonomous annotations across the cycle evolution in terms of incidence, overlap, increased accuracy, and the increment or stall of the pseudo-labeled annotations for each frame, as can be observed in Figure 5 (from left to right shows the progression during the self-supervised pseudo-labeling loop cycle). Moreover, it is important to notice the implementation of tracking techniques in order to increase the overall accuracy for each prediction, which interferes with an instance object detection analysis. Nevertheless, even though the collected annotations also presented a noisy dataset, some samples can be observed in Figure 6; the obtained metric results sustain the viability of the presented method for local model adaptation.

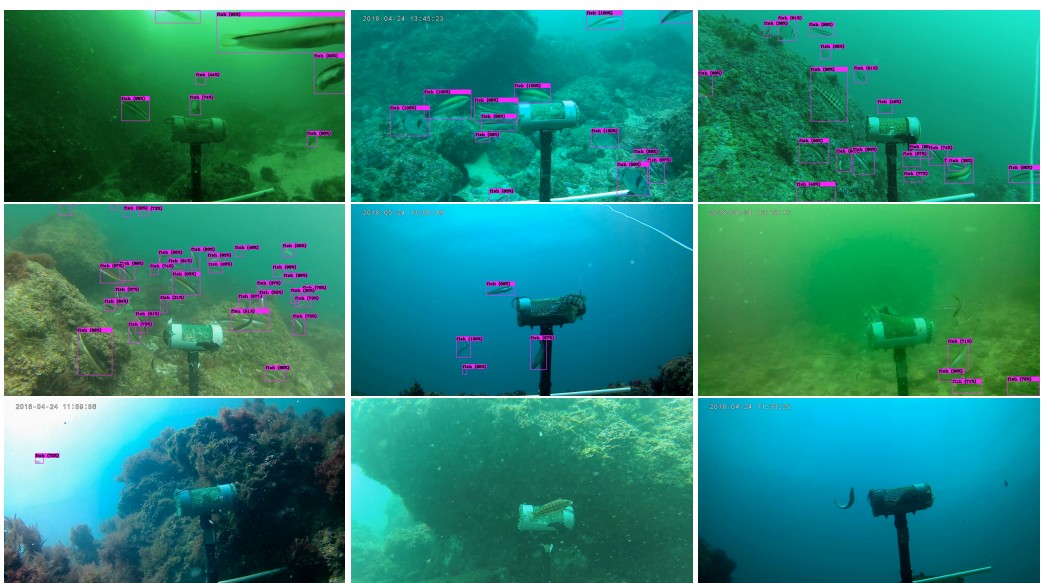

**Figure 6.** Sample of correct instant detections over frames obtained from the CCMAR footage. Top row: correct detections. Middle row: partially correct detections. Bottom row: failed cases.

Before proceeding to the final stage of the presented method, an EDA on the obtained pseudo-dataset was performed, which revealed different results relative to the underlying environmental conditions. While evaluating the bounding box automatic annotations, which influence the IoU metrics, a recurrent tendency across the fish movement was observed, where the end part and the tail zone were extended from the fins, leaving a blank space proportional to the size of the bounding box, while the head of the fish presented the opposite, with the head presenting itself on the edge or outside the bounding box annotation; some of these results can be reviewed in Figure 6. This effect can be the result of compounded errors of the applied tracking system during the training loop, or derivative of the rapid movement of the fish's tail.

To evaluate the automatically generated CCMAR dataset, we trained multiple models, varying the number of images from 4100 to 12,500 and 35,510, and afterward, the network size from $416 \times 416$ to $512 \times 512$ and $608 \times 608$, while training with 35,510 images. It is important to note that the amount of annotated images corresponds to half of the data used for training, and the other half represents the *negative sample*. Another relevant point is the almost proportional rise in the number of annotations versus the number of images, although the distribution of the number of fishes varied between one and more than sixty fishes in one frame. The images without annotations were not considered due to their possibility of containing false-negative detections. Moreover, the obtained metrics are presented in Table 1.

For the dataset size variation, an expected rise across all the relevant metrics was observed. The $mAP_{50}$ rose 4.66% from 88.33% when increasing the automated pseudo-labeled images from 4100, i.e., 2050 annotated images plus the *negative sample* to 12,500, having increased an insignificant 0.12% when expanding the number of images to 35,510. Similarly, the $mAP_{50}$ rose by 8.71% from 65.78%, and then by 0.21%, respectively. The IoU increase also followed similar proportions, rising 3.08% from 70.36%, and afterwards 1.66%. Precision rose from 85% to 87%, and finally to 88%, while recall increased from 84% to 86%, and then regressed to 84%.

Considering the obtained results, it was observed what apparently could become a threshold to predict the exact amount of images needed for a dataset, reducing the training requirements needed to obtain the desired results. Furthermore, the first sample increment introduced significant gains, while the latter even reduced the number of targets detected. Nevertheless, it is important to stress that increasing indiscriminately the amount of images from the presented method will introduce additional noise, similar to what can be observed

in Figure 6, although, in comparison with the OzFish dataset metrics, there was a significant increase across almost all metrics, specifically the $mAP_{75}$ and the recall. Notwithstanding, the obtained results are evaluated against the generated dataset at this stage; therefore, this analysis of these metrics must be evaluated in between the same data; however, a formal ground-truth evaluation is presented in Table 2.

**Table 2.** Results obtained from testing the models with the marine biologist-annotated images from different locations.

| Method | Dataset | Size | $mAP_{50}$ | $mAP_{75}$ | $\overline{IoU}$ [1] | Precision [1] | Recall [1] |
|---|---|---|---|---|---|---|---|
| YOLOv4 | OzFish | 608 | 48.60% | 27.56% | 76.22% | 94% \| 70% | 35% \| 44% |
| YOLOv4 | CCMAR | 608 | 74.05% | 43.40% | 72.29% | 90% \| 85% | 49% \| 66% |
| YOLOv4-tiny-3l | CCMAR | 608 | 68.62% | 38.30% | 78.83% | 98% \| 79% | 39% \| 61% |
| YOLOv4-tiny | CCMAR | 416 | 61.83% | 28.38% | 72.63% | 92% \| 73% | 38% \| 64% |

[1] For IoU threshold at 50%. Confidence threshold at 50% on the left, and 10% on the right.

Evaluating the effects of network input image size variation, an increase was observed in the $mAP_{50}$ of 0.84% from 91.90%, followed by an increase of 0.37%, while the $mAP_{75}$ rose 3.71% from 68.96%, and afterwards 2.03%. The variation of the IoU at 50% was also minimal, increasing 0.97% from 73.31%, and 0.82%. The precision only increased 1% from 88% when changing the network size from $512 \times 512$ to $608 \times 608$, while the recall also increased 1% from 83% when the network varied from $416 \times 416$ to $512 \times 512$. The number of images used for this evaluation was 35,510, as it had provided the higher results previously.

The obtained results demonstrate a small decay in mAP and IoU while reducing the network input image size, which confirms the potential of the presented method to automatically generate a robust and efficient distilled dataset adapted for the deployment of smaller networks, such as with edge devices. Therefore, a smaller version of the YOLOv4 was trained, the YOLOv4-tiny, with two YOLO layers, and the YOLOv4-tiny-3l, with the original three YOLO layers. In comparison, the YOLOv4 model has 137 pre-trained convolutional layers, while the YOLOv4-tiny has 29 pre-trained convolutional layers. The network image size varies from 416 for the smaller network to 608 for the head with three YOLO layers.

Using the same amount of images, a $mAP_{50}$ of 85.00% and 89.66%, respectively, was observed, as well as a significant increase in the $mAP_{75}$, with 45.33% and 61.02%. The IoU at 50% varied from 68.74% to 73.69%, while the precision varied from 87% to 90%, and the recall varied from 71% to 73%. Comparing these results to the obtained while using the complete YOLOv4 complete model, there is a decay in the performance at higher IoU, which is also expressed in a lower recall. Nevertheless, these outcomes have proven the viability of the proposed method for smaller neural networks aimed at edge deployment, especially considering the high $mAP_{50}$.

To further evaluate the performance and adaptive generalization of the proposed method, the trained models were evaluated on the unseen image collection annotated by a CCMAR marine biologist; these results are presented in Table 2. The first two models were trained with a network input image size of 608 and the full YOLOv4 model layers, with the first trained on the OzFish dataset and the second on the automatically generated CCMAR dataset. Comparing the two models, a significant increase in the mAPs was observed, with the $mAP_{50}$ rising 25.45% from 48.60% to 74.05%, and the $mAP_{75}$ increasing 15.84% from 27.56% to 43.40%. Curiously, the IoU decreased 3.93% from 76.22%; the same also occurred to precision, decreasing 4% from 94%. However, the recall increased 14% from 35% to 49%.

These results confirm a higher accuracy using the proposed method with a baseline noisy dataset to refit and automatically generate a new dataset directly associated with the desired final application. The lower IoU can derive from the observed issue of an additional annotation space after the fish's tail. The high precision obtained with the OzFish-trained model is directly correlated to the lower recall, meaning that the lower quantity of detected objects were the targets, while the CCMAR-trained model presented higher false positives.

It is possible to observe in Table 2, the direct relation between precision and recall with the variation of the confidence threshold.

The smaller network models evaluation on unseen images presented exceptional results, with the three YOLO-layered heads obtaining a $mAP_{50}$ of 68.62%, and a $mAP_{75}$ of 38.30%. The IoU increased in comparison with the previous validation, as presented in Table 1, to 78.83%, including the precision at 98%, although the recall majorly decreased to 39% for a confidence threshold of 50%. After evaluating the reason for such a low recall, two occurrences were detected: the smaller fish being undetected, which can be a result of the use of the YOLO architecture, as was previously mentioned, and a repetitive failure of detecting fishes surrounding the feeder, which also occurred with the training images, as can be observed on the bottom row of Figure 6. The smaller model also reproduced desired outcomes, with the $mAP_{50}$ of 61.83%, and with the $mAP_{75}$ of 28.38%, the IoU of 72.63%, the precision at 92%, and the recall at 38%, for a confidence threshold of 50%. Nevertheless, these results, specifically with high precision, can be deployed for highly accurate continual underwater station monitoring.

## 7. Conclusions and Future Work

The presented method allowed for an autonomous annotation of almost 700 min of raw footage; moreover, the distillation and augmentation techniques inside the self-learning loop significantly increased the detection rate, while also preventing the increase of false positives and false negatives. Additionally, multiple underwater environments and conditions, including obscured scenarios with the suspension of sand, were used and tested, proving its versatility. Furthermore, the obtained results against the ground-truth annotations of unseen images provided an exceptional instant detection mean average precision, even in smaller models aimed at edge deployment.

Future work is presented in Figure 7. It consists of the implementation of an advanced and more state-of-the-art tracking method, which will allow for more precise detection, either inside the self-supervising loop or when performing distillation while generating a new dataset. As for future dataset publication and availability, a more advanced tracking method allows for curating the more relevant images and sending them already pseudo-labeled for validation with the marine biologist, assisting in the creation of a ground-truth dataset. With the obtained results from the smaller networks, a possible implementation of inline training after deployment for in-site adaptation will be explored. Lastly, the development of a universal metric is mandatory for the evaluation of the progress during the self-supervising loop, and the application of the proposed method to different datasets and the cross-validation of the achieved results with the state of the art.

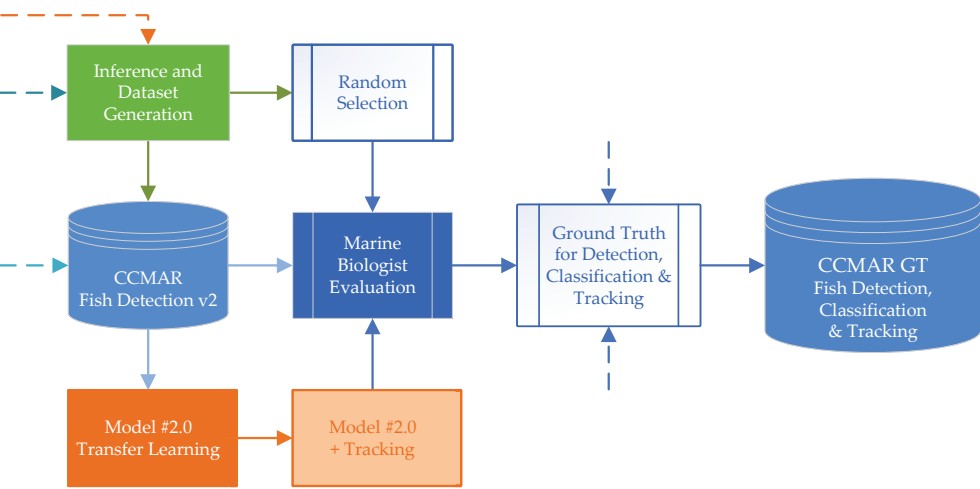

**Figure 7.** Future work.

**Author Contributions:** Conceptualization, R.J.M.V.; Data curation, R.J.M.V. and I.E.O.; Formal analysis, R.J.M.V.; Funding acquisition, J.S.; Investigation, R.J.M.V., J.P.S. and A.B.; Methodology, R.J.M.V.; Project administration, J.S.; Resources, L.B.; Software, R.J.M.V.; Supervision, J.S. and J.M.F.R.; Validation, R.J.M.V.; Visualization, R.J.M.V.; Writing—original draft, R.J.M.V.; Writing—review & editing, R.J.M.V., J.M.F.R. and L.B. All authors have read and agreed to the published version of the manuscript.

**Funding:** This work was financed by the project KTTSeaDrones for aerial and aquatic vehicles knowledge and technology transfer for the transboundary development of marine sciences and fishery — reference 0622_KTTSEADRONES_5_E—financed by the European Regional Development Fund (ERDF), POCTEP—Interreg VA Spain–Portugal, a program for cross-border cooperation. Also by the PO SEUR program (Operational Program for Sustainability and Efficient Use of Resources; Portugal 2020 Strategy) as the main financial support of the MARSW project (POSEUR-03-2215-FC-000046), and Portuguese national funds from FCT—Foundation for Science and Technology through projects UIDB/04326/2020, UIDP/04326/2020, UIDB/50009/2020 and LA/P/0101/2020. A.B. was supported by a doctoral grant from FCT (UI/BD/151307/2021).

**Institutional Review Board Statement:** Not applicable.

**Informed Consent Statement:** Not applicable.

**Data Availability Statement:** Not applicable.

**Acknowledgments:** CCMAR—Centro de Ciências do Mar—for their support, availability, and access to their data and knowledge. This work was supported by Portuguese Foundation for Science and Technology (FCT), LARSyS FCT-project UIDB/50009/2020, and CCMAR FCT-project UIDB/04326/2020.

**Conflicts of Interest:** The authors declare no conflict of interest.

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
