# Peer review of "Autonomous Temporal Pseudo-Labeling for Fish Detection"

_applsci, doi:10.3390/app12125910_

Round 1

Reviewer 1 Report

There is a strange use of commas in the first three pages of the paper - need to run through a grammar check to validate the use of commas.

This paper is a very interesting approach and a good application of transfer learning. Overall metrics for the system are important since using DL to generate labels could insert a situation that is difficult to interpret. It may be good to have some type of feedback with the SME to validate the automated labeling if not all of the processing needs to be done inside the underwater station.

Reviewer 2 Report

The authors present an exciting application. The use of real data is also valuable. My comments are:

1. The recall values are low; this is not fatal; however, a detailed explanation of why it was impossible to obtain a more significant value will be insightful. Also, establish why this low value does not affect your application.

2. Would you consider providing a Github with the datasets and code to reply to your results?

Round 2

Reviewer 2 Report

Thanks for addressing all the comments.